# Harvesting Design by Capital Return

**Petri P. Kärenlampi** 

Faculty of Science, University of Eastern Finland, P.O. Box 111, FIN-80101 Joensuu, Finland;
petri.karenlampi@uef.fi; Tel.: +358-50-371-1851; Fax: +358-13-251-4422

**Abstract:** One can suspect that capital return rate in forestry can be maximized by growing trees experiencing a value-adding assortment transition. Such a situation may or may not endure. We investigate the financial feasibility of a few harvesting schedules for a semi-shade-tolerant tree species. Some example stands have experienced commercial low thinning, whereas others have experienced young stand cleaning only. High thinning is computationally combined with quality thinning, and further growth of trees is estimated using an applicable growth model. High capital return rates are gained by diameter-limit cutting to the transition diameter between pulpwood and sawlogs. Repeated thinnings lead to a reduction in the capitalization during several decades, the system approaching a stationary state. The transient forest stands investigated show a significant excess capital return, in relation to the stationary state, and this excess return is due to transient tree size distribution. Correspondingly, capital return rate gained in rotation forestry is somewhat higher than that of stationary continuous-cover forestry, and the volumetric yield is much higher. The productive capacity of stands previously thinned from below has been apparently ruined by that treatment.

**Keywords:** assortment transition; diameter-limit cutting; improvement harvesting

## 1. Introduction

One can intuitively suspect that capital return rate in forestry can be maximized by growing trees experiencing a value-adding assortment transition. The volumetric growth within a stand can possibly be allocated to trees experiencing such an assortment transition by harvesting trees that already have completed most of the transition. Such a transition situation may possibly endure over several harvesting cycles, provided the tree species is at least somewhat shade-tolerant.

There is a large body of literature discussing the effect of thinning schedules on the net present value of revenues [1–9]. It appears that the optimal number of thinnings, thinning intensity, as well as selection between continuous-cover forestry and clearcuttings depends on the applied discounting interest rate [1–9]. The discounting interest rate is supposed to reflect the cost of capital; however, the actual expense of borrowing varies widely, and the opportunity costs in terms of missed alternative investments tend to vary more. We find such a discounting approach inappropriate, since consequences may be financially devastating [10].

An improvement harvesting may contribute to the capital return rate on a forest stand. We intend to design such an initial improvement harvesting. Provided significant improvement of capital return rate is possible through harvesting, we look forward to further harvestings in order to retain a high capital return rate. We take it granted that removal of trees with obvious quality defects or visible loss of vigor improves further productivity of the stand. In addition to the quality thinning, the effect of diameter-limit cutting on the capital return rate is investigated.

We adopt a practical focus to a few fertile Norway spruce-dominated stands in Eastern Finland, where observations regarding the present structure of the tree populations are collected. Then,

the growth of trees, as well as recruitment of new trees, is clarified according to an empirical growth model, constructed on the basis of a large Norwegian dataset [11,12]. Even if the practical focus is in semiboreal spruce stands, the results may be, at least qualitatively, applicable to any circumstances where at least to some degree shade-tolerant trees experience a value-adding assortment transition. Different tree species, however, are likely to require a calibrated growth model of their own.

First, materials and methods are introduced. This offers a description of the example sites, the applied growth model, as well as financial methods. Then quality thinning is introduced on the basis of quality classification of the trees recorded in field measurements, and the quality thinning is combined with diameter-limit cutting. The capital return rate is reported as relative value growth rate within a five-year period after any thinning. The capital return rate is clarified first after quality thinning only, and then as a function of any applied cutting limit diameter. Finally, simulated further development of the stands under repeated diameter-limit cuttings is reported, and some implications of the results are discussed.

## 2. Materials and Methods

### 2.1. Experimental Materials

Eleven circular plots of an area of 314 m$^2$ were taken from typical spots of eleven spruce-dominated forest stands in November 2018 at Vihtari, Eastern Finland. Seven of the stands had experienced only young stand cleaning, whereas four of the stands were previously thinned commercially. The occurrence of a previous thinning was determined on the bases of observable striproads on the stands. Breast-height diameters were recorded, as well as tree species, and a quality class was visually determined for any measured tree.

Figure 1 shows basal area of trees at breast height and stem count per hectare on experimental plots taken from the eleven example stands. The rightmost marker on any curve reports all the trees appearing on the plots. The other markers in the Figure report only trees determined to be of acceptable quality for growing further. Of these, the leftmost marker reports trees of breast-height diameter of at most 150 mm. The second from left reports trees of breast-height diameter of, at most, 200 mm. The markers further right report trees of progressively higher maximum diameter, at 50 mm intervals.

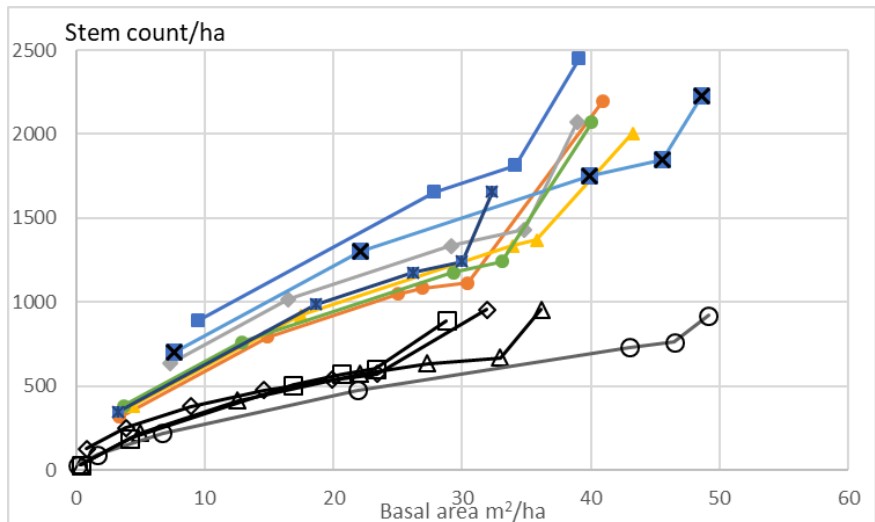

**Figure 1.** Stem count and basal area of seven stands not previously thinned commercially (filled markers), and four stands thinned commercially (unfilled markers). The rightmost marker within any curve includes all trees on the measured plot. Other markers correspond only trees determined to be of acceptable quality for further growing. The leftmost marker within any curve corresponds to trees not larger than 150 mm in breast-height diameter. The diameter limit is increased by 50 mm for each step to the right.

Figure 1 shows that measured plots on sites that had not experienced any commercial thinning were relatively similar to each other: basal area varied from 32 to 48 m$^2$/ha, and the stem count from 1655 to 2451 per hectare (the rightmost filled marker on any curve in Figure 1).

Figure 1 also shows that measured plots on stands previously thinned commercially also were relatively similar to each other: basal area varied from 29 to 49 m$^2$/ha, and the stem count from 891 to 955 per hectare (rightmost unfilled marker on any curve in Figure 1). Again, the leftmost marker of any curve in Figure 1 corresponds to the basal area and stem count of good-quality trees of at most 150 mm of breast-height diameter, and further markers in any curve correspond to basal area and stem count of good trees of at most 200, 250, 300, 350, and 400 mm of diameter. Correspondingly Figure 2 indicates that the measured plots did not contain any trees thicker than 400 mm. We also find that the stem counts in the smallest diameter classes in Figure 2 are rather low, which indicates the sites have experienced thinning from below.

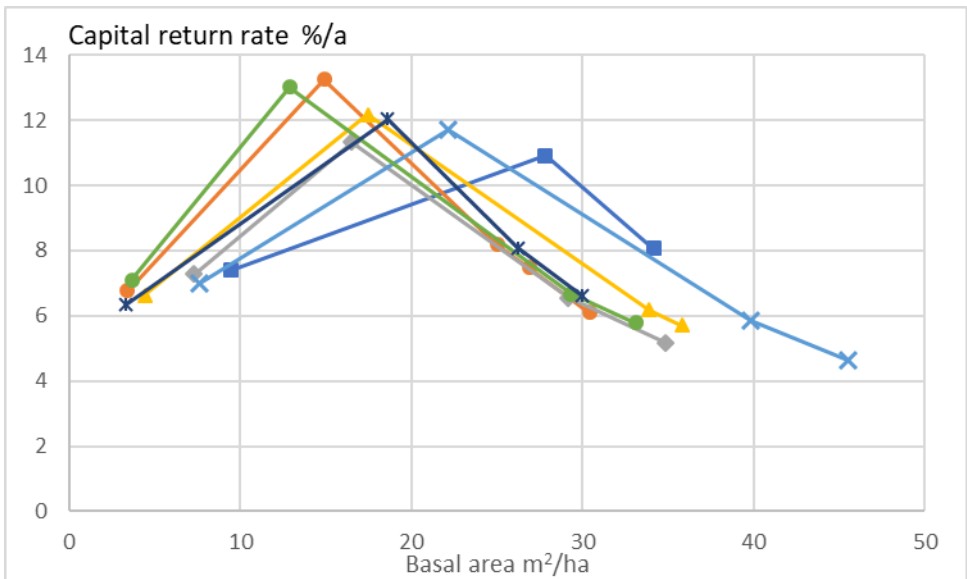

**Figure 2.** Capital return rate and basal area of seven stands not previously thinned commercially. The rightmost marker within any curve includes all trees determined to be of acceptable quality for further growing. The leftmost marker within any curve corresponds to trees no larger than 150 mm in breast-height diameter. The diameter limit is increased by 50 mm for each step to the right.

*2.2. Applied Growth Model*

In order to discuss further development of the example stands, represented by the measured circular plots, some kind of a growth model is needed. The growth model of Bollandsås et al. [11,12] is adopted, discussing not only growth but also mortality and recruitment. In the case of applications discussed in this paper, growth, and recruitment are essential but the role of mortality is small. Trees are discussed in diameter classes of 50 mm, and each class is represented by its central tree. Growth is computationally implemented for any five-year period in terms of the probability of a tree to transfer to the next diameter class [11–13]. Similarly, for any five-year period, there is a computed mortality within any diameter class, and an expected value of recruitment into the smallest diameter class [11–13]. Any five-year development depends on the initial condition. The initial condition for any five-year period is to be described below.

*2.3. Financial Valuations and Methods*

In order to discuss financial issues, we need to address value growth, instead of merely volumetric growth. For any diameter class, the volumetric amount of two assortments, pulpwood and sawlogs, is clarified according to an appendix given by of Rämö and Tahvonen [14,15]. The monetary value of

the assortment volumes is taken as national stumpage prices from year 2000 to 2011 as given by Rämö and Tahvonen [14].

To determine a momentary capital return rate, or possibly an average capital return rate for a period of a few years, we need to discuss the amount of financial resources occupied [10,13]. This is done in terms of a financial potential function, defined in terms of capitalization per unit area *K*. The momentary capital return rate becomes:

$$r(t) = \frac{d\kappa}{K(t)dt},$$　　　　　　　　　　(1)

In Equation (1), the difference between $\kappa$ in the numerator and *K* in the denominator relates to eventual operative investment or divestment. The potential (or capitalization) *K* is immediately affected by any eventual operative investment or withdrawal, and then consequently becomes affected by amortizations. The net return rate $\frac{d\kappa}{dt}$ in the numerator is not immediately affected by investments or withdrawals, but considers eventual investments in terms of amortizations. In addition, investments are likely to contribute to growth: they probably increase growth rate, whereas withdrawals may reduce growth rate. In this paper, we do not discuss investments, amortizations or withdrawals during growth periods where Equation (1) is applied. Correspondingly, the capitalization in the denominator of Equation (1) contains value growth of trees and the value of the bare land, and no other elements.

Let us then distribute capitalization *K*(*t*) to operative capitalization *O*(*t*), and non-operative capitalization *U*(*t*) [10,13]. The operative capitalization relates to cumulated growth, as well as eventual investments related to it. Non-operative capitalization may be due to excess demand of real estate, in comparison to supply, recreational values, speculation for future real estate development, etc. Now, Equation (1) can be rewritten:

$$r(t) = \frac{d\Omega + dU}{[O(t) + U(t)]dt},$$　　　　　　　　　　(2)

In Equation (2), the difference between $\Omega$ in the numerator and *O* in the denominator again relates to eventual operative investment or divestment. In this paper, however, we do not discuss investments, amortizations or withdrawals during growth periods where Equation (2) is applied. This practically equates $\Omega$ and *O* in the present application.

Equation (2) reveals that in the case the operative capitalization is much higher than the non-operative capitalization, the role of the latter vanishes. On the other hand, if non-operative capitalization is much higher than operative capitalization, the role of the operative capitalization vanishes. In case the non-operative capitalization is large but constant, the highest operative return might simply be the one corresponding to the greatest average yield rate $\left\langle \frac{d\Omega}{dt} \right\rangle$. The situation is somewhat more delicate if there is a nonvanishing time change rate of the non-operative capitalization $dU/dt$.

In general, we include bare land value in the non-operative capitalization *U*. In this paper, we discuss the bare land value as the only component of the non-operative capitalization. It is assigned a present value of 600 Euros/ha, and an annual appreciation of 3%. Both of these values correspond to local circumstances at Vihtari, Eastern Finland, at the time of writing.

## 3. Results

### *3.1. Stands Not Previously Thinned Commercially*

Annualized capital return rate within a five-year period after improvement harvesting, as a function of basal area, in the case of stands not previously thinned commercially, is shown in Figure 2. Again, the leftmost marker of any curve corresponds to the basal area of and capital return from trees of at most 150 mm of breast-height diameter. Further markers in any curve correspond to basal area and capital return from trees of at most 200, 250, 300, and 350 mm of diameter. We find that without

any exception, the greatest capital return rate is gained if there is, in addition to the removal of trees of low quality, a diameter-limit cutting to 200 mm. Such stands, after the improvement harvesting, deliver annual capital return rates from 10.9% to 13.2%. Diameter-limit cutting to 150 mm would give a capital return rate in the vicinity of 7%, which, in general, is greater than that achievable with gentle quality thinning only.

A stand experiencing periodic diameter-limit cuttings after the improvement harvesting will not be in any (fluctuating) stationary state. Annualized capital return rate does evolve along with further growth and diameter-limit cuttings, implemented any five years after the improvement harvesting. We find from Figure 3 that all stands however do approach a stationary state within a century, the stationary capital return rate approaching 8% (Figure 3). There is some variation in the stationary capital return rate, due to somewhat varying site fertility. It is worth noting that all the capital return rates in Figure 3 correspond to present time. In other words, the bare land value, as well as its appreciation rate, are taken as the present values. In other words, in Figure 3, the improvement harvesting is assumed to have occurred in variably distant history.

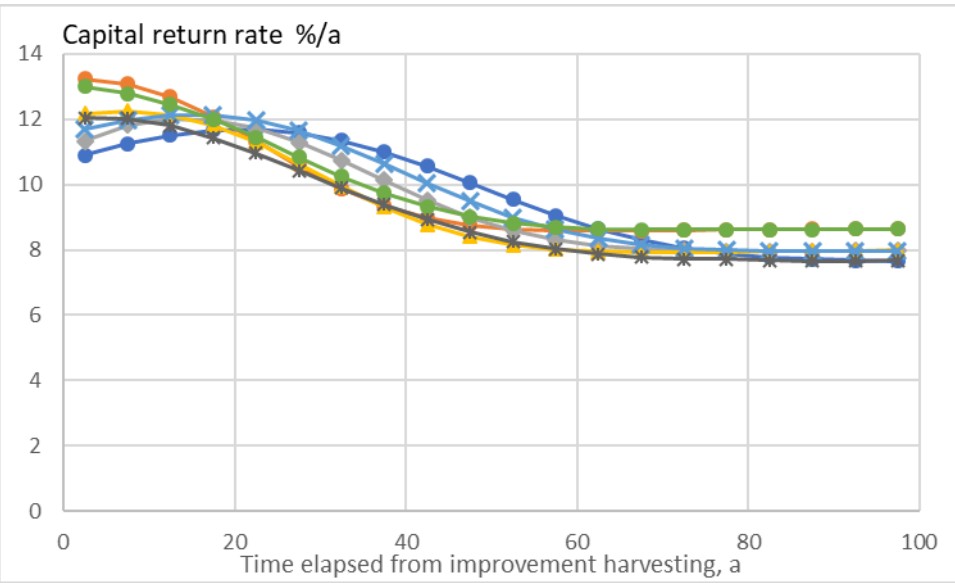

**Figure 3.** Development of capital return rate within seven stands not previously thinned commercially. The leftmost point corresponds to the first five-year period after the improvement harvesting. Along with repeated diameter-limit cuttings, all stands approach a stationary state.

There is a significant excess capital return during the first decades after the improvement harvesting, in relation to the stationary state (Figure 3). This is due to tree size distributions significantly differing from the stationary state. Some stands show a maximum in the capital return rate immediately after the improvement harvesting. These stands have a mode value of the tree size distribution close to the transition from pulpwood to sawlogs. Some other stands show a maximum in capital return 5–25 years after the improvement harvesting. These stands have a mode value of tree sizes below the transition diameter, and the capital return rate increases as the number of trees approaching the transition diameter increases (Figure 3).

The experimental stands not being in any stationary state after the improvement harvesting, the capital return rate is not the only one of their features experiencing a transient. It is found from Figures 4 and 5 that the basal area after any diameter-limit cutting, as well as the stem count per hectare, however do approach a stationary state. The stationary values of basal area and stem count are rather low, 2.2 $m^2$/ha and 170/ha, such low values being related to relatively slow recruitment of new trees [11–13].

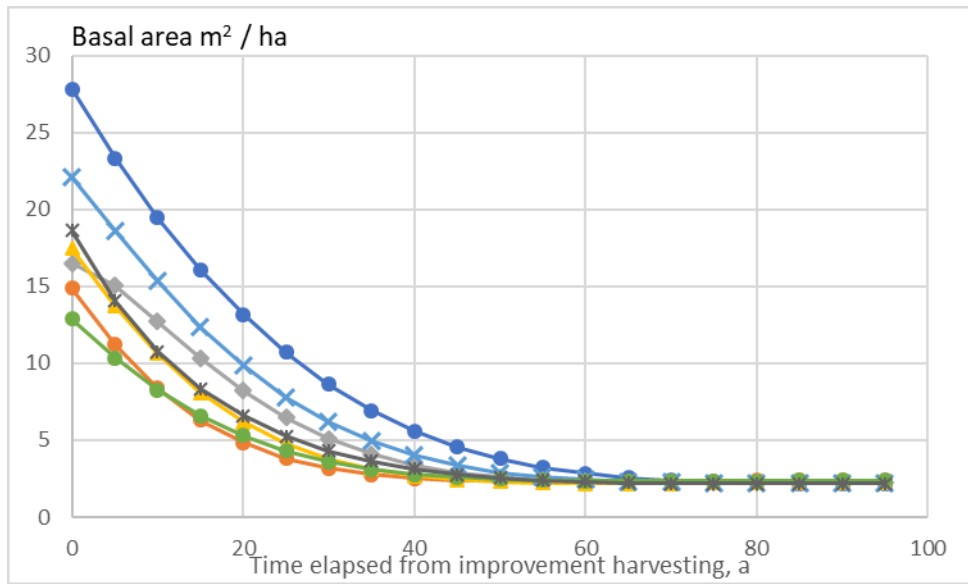

**Figure 4.** Development of basal area within seven stands not previously thinned commercially. The leftmost point corresponds to the situation immediately after the improvement harvesting. Along with repeated diameter-limit cuttings, all stands approach a stationary state.

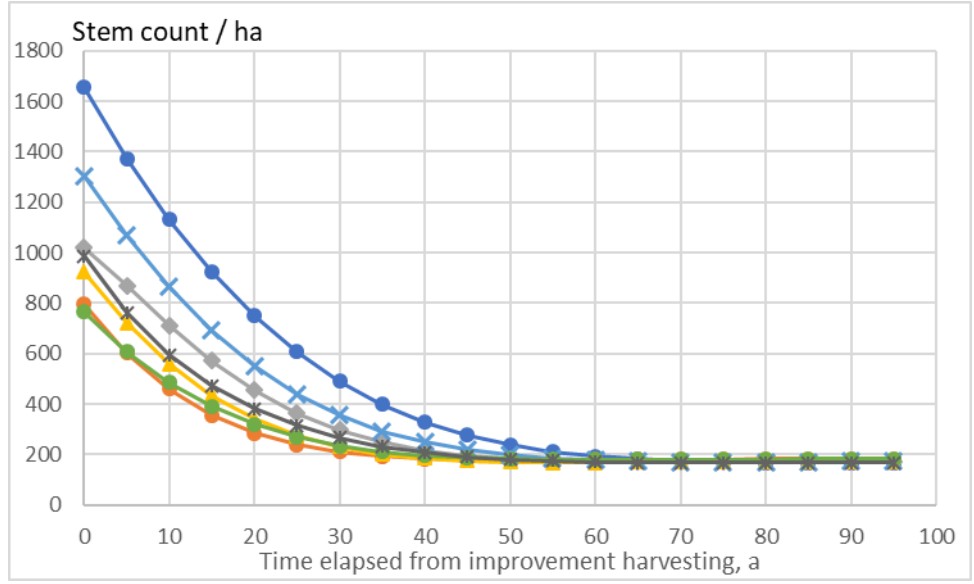

**Figure 5.** Development of stem count within seven stands not previously thinned commercially. The leftmost point corresponds to the situation immediately after the improvement harvesting. Along with repeated diameter-limit cuttings, all stands approach a stationary state.

Figures 4 and 5 show that during the first decades after the improvement harvesting, there generally are significant stem counts and basal areas after any diameter-limit cutting on the example sites without previous commercial thinning. However, there also is significant variability between the stands. Before any harvesting, most of the basal areas have been in the vicinity of 40 m$^2$/ha (Figure 1), and most of the basal area has been removed in the improvement harvesting (Figure 4). However, in the case of two stands, a significant amount of basal area has remained (Figure 4). This obviously is related to the large remaining stem count visible in Figure 5: these two stands have had a large number of trees smaller than 200 mm in breast-height diameter.

As there is a significant excess in basal area and stem count during the first decades after improvement harvesting, in comparison to the stationary state, there also is an excess in the harvesting

yield (Figure 6). In the stationary state, the harvesting yield is in the order of 7 m$^3$/ha for any five-year period, and the excess yield during the first decades after the improvement harvesting is rather significant. There is one example stand where the change in the harvest volume is non-monotonic. In that case, the mode value of tree size was well below the pulpwood-sawlogs transition size after the initial improvement harvesting.

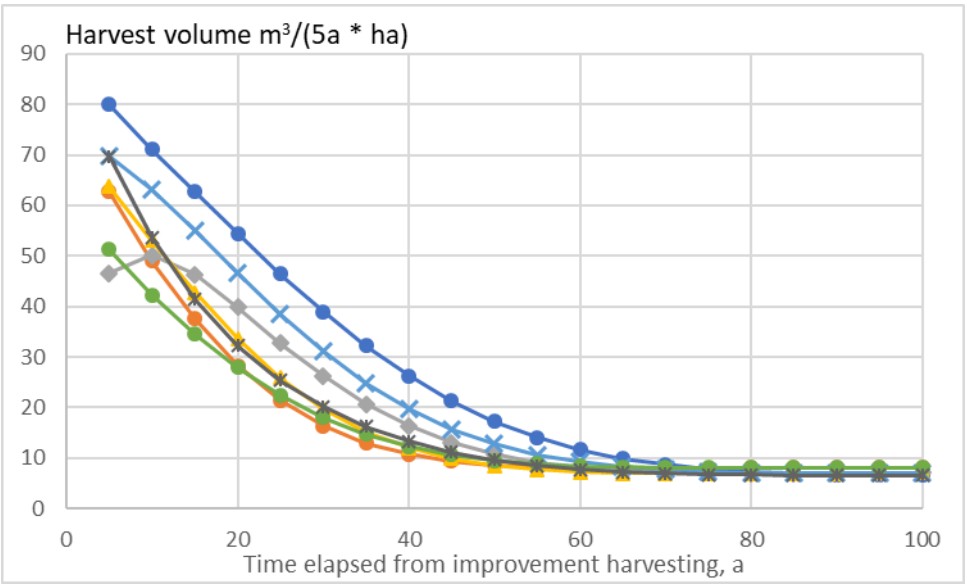

**Figure 6.** Development of harvest volume/ha within any five-year period within seven stands not previously thinned commercially. The leftmost point corresponds to the situation five years after the improvement harvesting. Along with repeated diameter-limit cuttings, all stands approach a stationary state.

It is worth noting that the five-year harvest yield 20 years after the improvement harvesting is in the order of 30 m$^3$/ha even if the basal area (after harvesting) is only 6 m$^2$/ha (Figures 4 and 6). The harvest yield mostly contains sawlogs.

*3.2. Stands Previously Thinned Commercially*

Annualized capital return rate within a five-year period after improvement harvesting, as a function of basal area, in the case of stands previously thinned commercially, is shown in Figure 7. Again, the leftmost marker of any curve in Figure 7 corresponds to the basal area of and capital return from trees of at most 150 mm of breast-height diameter. Further markers in any curve correspond to basal area and capital return from trees of at most 200, 250, 300, 350, and 400 mm of diameter. Again, it is found that without any exception, the greatest capital return rate is gained if there is, in addition to removal of trees of low quality, a diameter-limit thinning to 200 mm. Such stands, after the deliver annual capital return rates from 8.6% to 12.0%. Diameter-limit cutting to 150 mm would give capital return rate in the vicinity of 4%, which in general is greater than that achievable with gentle quality thinning only (Figure 7).

It is worth noting that the greatest capital return rates are achieved in stands where the basal area is only 1.7–5.0 m$^2$/ha (Figure 7). This is much less than in the case of the unthinned stands shown in Figure 2, and again indicates that the stands in Figure 7 have been thinned from below.

Again, any stand experiencing periodic diameter-limit cuttings after the improvement harvesting will not be in any (fluctuating) stationary state. Annualized capital return rate does evolve along with growth and further diameter-limit cuttings, implemented any five years after the improvement harvesting. We find from Figure 8 that all stands approach a stationary state within a century, the stationary capital return rate approaching 8%–9% (cf. Figure 3). There is some variation in

the stationary capital return rate, due to somewhat varying site fertility. Again, all the capital return rates in Figure 8 correspond to present time. In other words, the bare land value, as well as its appreciation rate, are taken as the present values. In other words, in Figure 8, the improvement harvesting is assumed to have occurred in variably distant history.

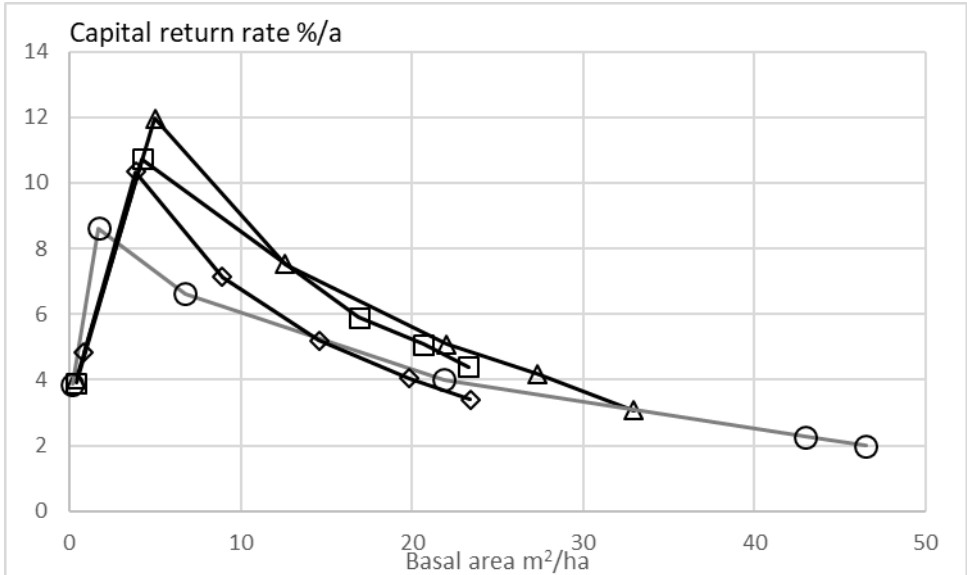

**Figure 7.** Capital return rate and basal area of four stands previously thinned commercially. The rightmost marker within any curve includes all trees determined to be of acceptable quality for further growing. The leftmost do within any curve corresponds to trees not larger than 150 mm in breast-height diameter. The diameter limit is increased by 50 mm for each step to the right.

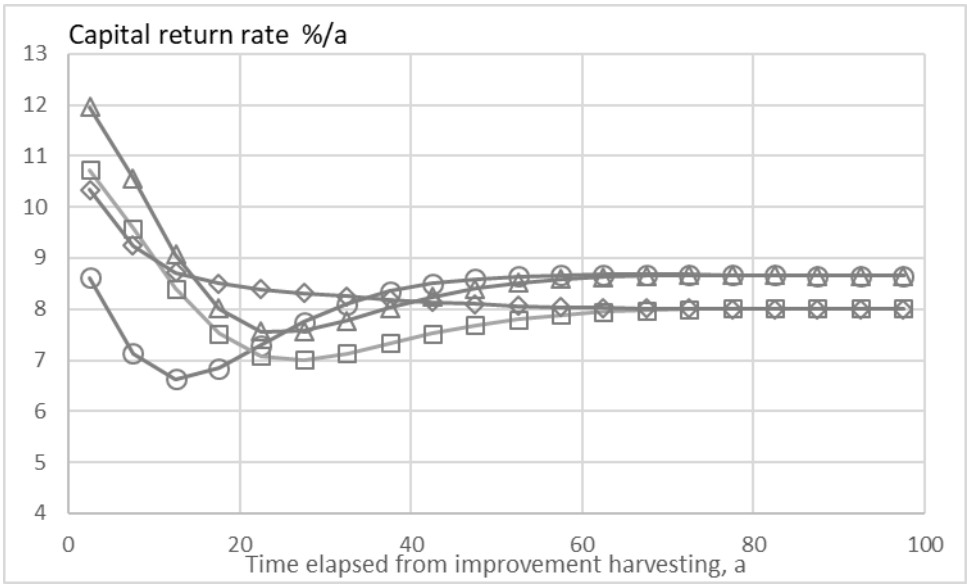

**Figure 8.** Development of capital return rate within four stands previously thinned commercially. The leftmost point corresponds to the first five-year period after the improvement harvesting. Along with repeated diameter-limit cuttings, all stands approach a stationary state.

There is an excess capital return during the early stages of development after the improvement harvesting, in relation to the stationary state (Figure 8), with one exception. In the case of one stand only, the development of the capital return rate is monotonic, like in the case of Figure 3. In three cases the capital return rate shows a depressed state 10–40 years after the improvement harvesting,

in comparison to the stationary state. This obviously is due to a low number of trunks experiencing the pulpwood-sawlog transition.

The experimental stands not being in any stationary state after the improvement harvesting, the capital return rate is not the only one of their features experiencing a transient. The basal area after any diameter-limit cutting, as well as the stem count per hectare, however, also approach a stationary state along with time. Only one of the four stands shows a monotonic decrement of the basal area and stem count. Three of the four stands show a depressed basal area and stem count after the improvement harvesting, in comparison to the stationary state. In other words, the behavior of basal area and stem count is qualitatively similar to the behavior of capital return rate in Figure 8. One of the three stands however shows a depressed basal area and stem count right after the improvement harvesting, even if the capital return rate is not depressed at that state (Figure 8).

All the four stands show an excess five-year harvest volume after the improvement harvesting, in relation to the stationary state (Figure 9). Again, the harvest volume decreases monotonically only in one case, the other three cases showing a depressed harvest volume 10–40 years after the improvement harvesting.

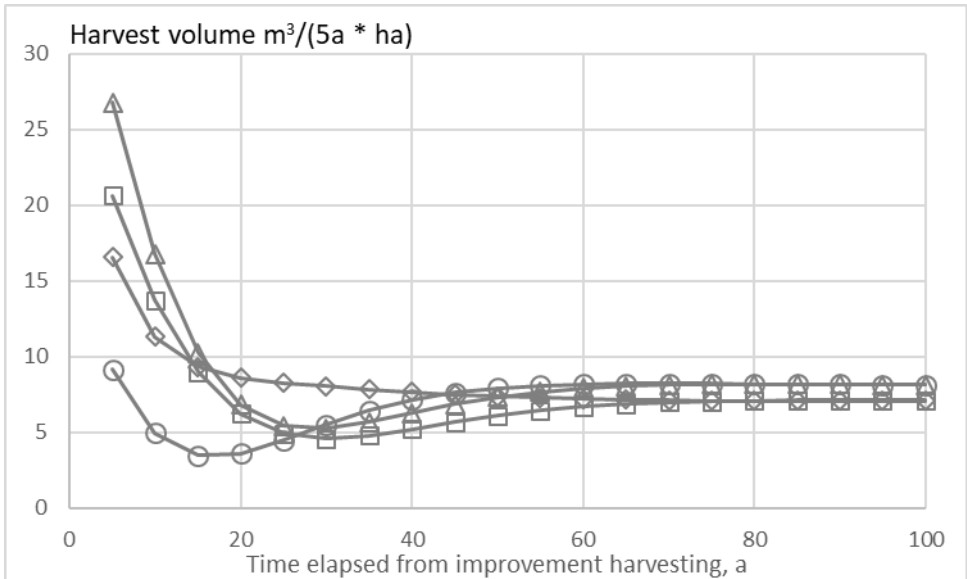

**Figure 9.** Development of harvest volume/ha within any five-year period within four stands previously thinned commercially. The leftmost point corresponds to the situation five years after the improvement harvesting. Along with repeated diameter-limit cuttings, all stands approach a stationary state.

## 4. Discussion

One must raise a question regarding the capital return rate over the entire rotation, including the decades necessary to grow a young stand to a state where the improvement harvesting can be implemented. The present data does provide some tools for solving this question. Firstly, the characteristic tree age of the seven non-thinned stands discussed varied from 31 to 45 years. Second, the computed stumpage value of trees to be collected in the improvement harvesting varied 2400–9200 Euros/ha. Third, the present bare land value, as well as the present level of artificial regeneration and young stand cleaning expenses are known. These can be discounted to the time of artificial regeneration and young stand cleaning. A natural discounting interest possibly equals the present bare land appreciation rate.

It has been recently shown that an accurate computation of a representative (expected) value of capital return rate requires knowledge of the details of the yield function [10]. However, an approximation can be gained simply as an internal rate of return [10]. Considering the numbers discussed in the preceding paragraph, the annual internal return rates up to the improvement

harvesting appear to be 3.7%–7.5%, with a mode value of 6.6%. This is less than the capital return rate in the stationary state (Figure 3). However, considering the fact that there is a significantly elevated capital return rate during the first four decades after the improvement harvesting (Figure 3), rotation forestry with repeated high thinnings apparently provides somewhat greater capital return rate as stationary continuous-cover forestry.

In the context of a comparison between rotation forestry with repeated high thinnings on the one hand and stationary continuous-cover forestry on the other, one possibly should discuss other factors, in addition to the capital return rate. Firstly, the harvest volume is much greater in rotation forestry (Figure 6). Secondly, the low basal area and stem count in the stationary state (Figures 4 and 5) may induce a significant risk of wind damage. An intense improvement harvesting in rotation forestry also may induce an elevated risk of wind damage [16,17], but that risk can be reduced by implementing the improvement harvesting in stages, if necessary. Thirdly, risks involved in lengthy stationary forestry in terms of diseases and eventual loss of vigor are not fully known.

Stands previously thinned commercially demonstrate a rather small basal area and stem count after the improvement harvesting (Figures 1 and 7). The scarcity of pulpwood-sized trees results as a rapid decline in basal area, stem count, harvest yield and capital return rate (Figures 8 and 9). On the other hand, a significant increment of capital return rate can be achieved by partial diameter-limit cutting, or by thinning from above (Figure 7). A larger cutting limit diameter obviously would increase the remaining basal area and harvest yield, at the expense of a lower capital return rate after the improvement harvesting (Figure 7).

Figure 10 shows the evolution of the capital return rate in previously thinned stands, but now with 250 mm cutting limit diameter. In comparison to Figure 8 we find that the situation is clearly impaired. The initial values of capital return rate, gained after the improvement harvesting, are much lower, as already indicated in Figure 7. The capital return rate in the stationary state also is lower with the higher cutting limit diameter (Figure 10).

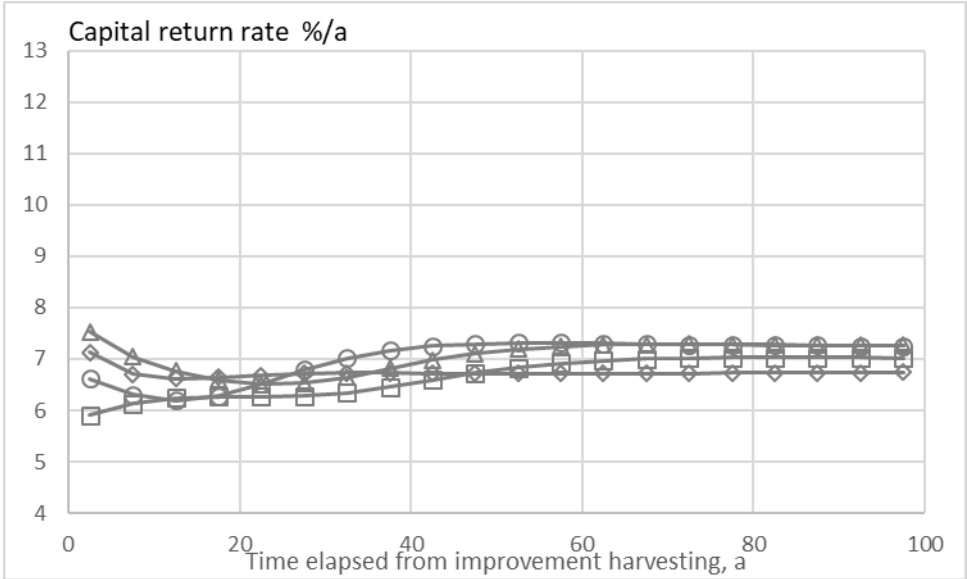

**Figure 10.** Development of capital return rate within four stands previously thinned commercially. The leftmost point corresponds to the first five-year period after the improvement harvesting with cutting limit diameter 250 mm. Along with repeated diameter-limit cuttings, all stands approach a stationary state.

There are some benefits that are achieved at the expense of the lower capital return rate. The stem count, and in particular the basal area is greater, and this corresponds to a greater volumetric harvest yield as is shown in Figure 11, in comparison to Figure 9. Not only the initial harvest volume,

gained five years after the improvement harvesting, is greater, but the stationary harvest volume also is greater. Even if the capital return rate in Figure 11 is clearly less than in Figure 9, it is much greater than in the absence of any diameter-limit cutting in Figure 7. This does, however, not change the fact that much of the financial productivity has been ruined by thinning from below, in comparison of Figure 10 to Figure 3.

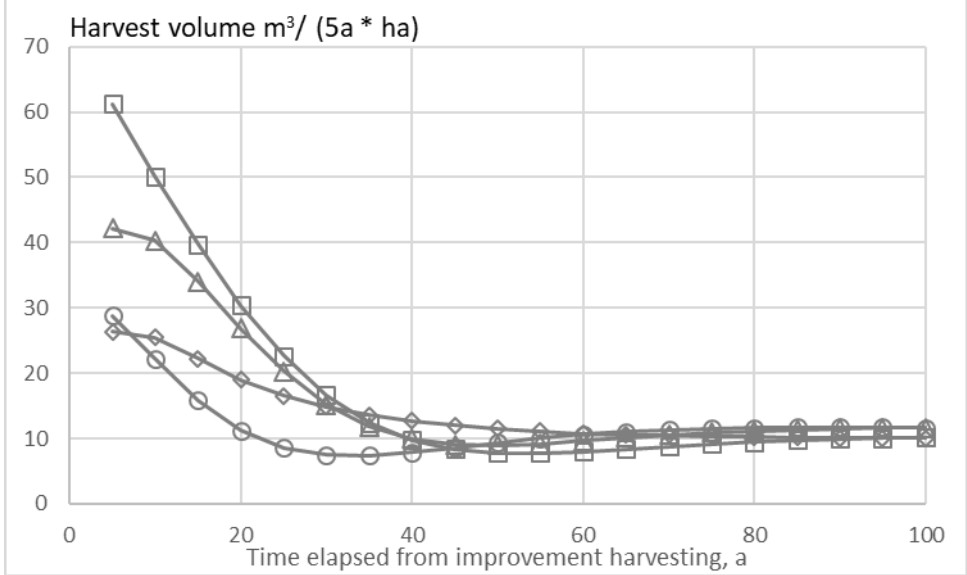

**Figure 11.** Development of harvest volume/ha within any five-year period within four stands previously thinned commercially. The leftmost point corresponds to the situation five years after the improvement harvesting with cutting limit diameter 250 mm. Along with repeated diameter-limit cuttings, all stands approach a stationary state.

Several earlier studies indicate stationary continuous-cover forestry would be more profitable than rotation forestry [2–4,8]. The results of this paper are contradictory. There are several obvious reasons for the discrepancy. Firstly, the economic criteria differ. Most commonly, a discounted net present value of revenues has been maximized, using an external discounting interest rate [2–4,8,18]. It has been recently shown that maximization of net present value in general yields incorrect results from the viewpoint of wealth accumulation, particularly in the case of fertile stands [10]. Secondly, stationary continuous-cover forestry has been compared with clearcuttings, instead of a procedure of repeated high thinnings [2,3]. It is also worth noting that sustainability has been secured by using a penalty function for declining stem counts [3]. Declining stem counts however are an essential feature of Figure 5, and it is closely related to the excess capital return appearing in Figure 3.

Interestingly, the cutting programs of high capital return display many of the traits demonstrated as optimal by Kilkki and Väisänen [1], even if their objective function was a discounted net present value of revenues. A possible explanation for the similarity of the procedures is that the applied discounting rate accidentally was not far from the internal rate of return achievable in their example stands [1,10].

There are uncertainties in the growth model applied. The most uncertain submodel within the experimental model applied here is the recruitment model [11]. The recruitment model is crucial from the viewpoint of the amount of trees that can be maintained at the stationary state. The recruitment model is less essential in the case of rotation forestry containing artificial regeneration, improvement harvesting, and repeated high thinnings after the improvement harvesting.

It would be of interest to compare the present outcome with different growth modellings, as well as with different tree species, climates, and regions. Quite a few investigations have been published reporting recruitment, growth, and mortality [5,19–23]. However, it appears that most of

such modellings do not converge to any natural stationary state under a demographic stationarity criterion [13]. A common reason for such failure appears to be an inappropriate description of mortality: in case growth rate diminishes but mortality does not increase, a large number of trees accumulates to large diameter classes. It also would be beneficial to compare with Asian and American tree species. Excellent field studies are available [24,25], but well-formulated growth models would be needed for detailed financial analysis.

It remains to be investigated whether various growth models would produce coherent results under the boundary condition of frequently repeated diameter-limit cuttings. However, we have implemented some preliminary investigations using the growth model of Pukkala et al. [19]. Firstly, the greatest capital return rate is gained by diameter-limit harvesting to the transition diameter between pulpwood and sawlogs, regardless of the growth model. However, the rate of recruitment is faster in the model of Pukkala et al. [19], resulting as basal area, stem count and harvest volume larger than indicated for the stationary state in Figures 4–6. Secondly, the effect of spacing on diameter growth is stronger according to the model of Pukkala et al. [19], resulting as high growth rates in sparse stands. On the other hand, dense stands hardly achieve a basal area of 45 m$^2$/ha within a century. Stands with basal area over 55 m$^2$/ha exist in the area, in agreement with the growth model of Bollandsås et al. [11].

In addition to tree species, growth conditions, and models describing growth and recruitment, financial and technical circumstances would vary by geographic region. The basic requirement for the applicability of the results of this study is that trees experience a value-adding assortment transition. The magnitude of the value transition may be a function of not only location but also time. Assortment values given by Rämö and Tahvonen [14] in terms of national stumpage prices from 2000–2011 correspond to the valuation of sawlogs in the example area during the time of writing rather accurately. However, the valuation of pulpwood in the example area is now about 15% less. This makes the value transition sharper than descried in the numerical treatments above. Correspondingly, the capital return rate becomes higher at the assortment transition, and it is even more essential to concentrate growth in trees experiencing the assortment transition. A less pronounced value transition, in comparison to values given by Rämö and Tahvonen [14], would have the opposite effect.

There are other issues contributing to the sharpness of the value transition, in addition to the percentage value increment from one assortment to another. The volumetric amount of two assortments, pulpwood and sawlogs, as a function of tree diameter, was given by of Rämö and Tahvonen [14,15] in terms of a deterministic array. In reality, the yield of sawlogs is stochastic, depending on the appearance probability of defects on the one hand, and tolerance of defects on the sawlog market on the other hand. Numerical treatments of this paper were based on a simplified description of any 50 mm diameter class by its central tree, and the first diameter class assumed to yield sawlogs was centered in 225 mm diameter. Considering the stochastic nature of the sawlog yield, the cutting limit diameter possibly should not be taken at the lower end of this diameter class, but is should be selected somewhere within this diameter class so that the probability of gaining a sawlog will become high.

It is evident from Figure 4 that high capital return rate is gained from stands with relatively low basal area of trees, and correspondingly relatively low standing trunk volume, as well a low capitalization per unit area. An increment of standing volume would increase growth, at the expense of capital return rate (compare Figures 9 and 11, and Figures 8 and 10). Stands of high capital return rate but low capitalization, and at most moderate volumetric growth are not effective in carbon sequestration [26,27]. We intend to discuss this in more detail in a forthcoming paper.

## 5. Conclusions

A significant increment in the capital return rate is gained by diameter-limit cutting to the transformation size of pulpwood to sawlogs, compared with quality thinning only (Figures 2 and 7). Apart from that, two rather different sets of results have been introduced.

Stands not previously thinned commercially provide a significant capital return rate, basal area, stem count, and five-year harvest yield during several decades after the improvement harvesting. In fact, these are in excess of the level of the stationary state for a half century (Figures 3–6).

Stands previously thinned commercially demonstrate a rather small basal area and stem count after the improvement harvesting (Figure 7). The scarcity of pulpwood-sized trees results as a rapid decline in basal area, stem count, harvest yield, and capital return rate (Figures 8 and 9). Such quantities even tend to stay in a depressed state, in comparison to the stationary state, for several decades (Figures 8 and 9). One can, without doubt, argue that the productive capacity of these stands has been ruined by thinning from below.

Young previously unthinned stands show significant excess financial return after the improvement harvesting, in comparison to the stationary case, during several decades (Figure 3). This, combined with excess harvest volume (Figure 6), possibly suggests superiority of rotation forestry with repeated high thinnings, in comparison to stationary continuous-cover forestry.

**Acknowledgments:** In December 2018, a partial improvement harvesting has been implemented on the example stands discussed in this paper. The improvement harvesting has been partial in order to avoid wind and snow damage, and it will be completed within a few years. Aside from practical forestry activities, the author does not have any particular interest to declare in relation to this paper.

**Conflicts of Interest:** The author declares no conflict of interest.

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
