# Peer review of "Harvesting Design by Capital Return"

_forests, doi:10.3390/f10030283_

Round 1
Reviewer 1 Report
a) General comment :
The authors present a study to evaluate the financial feasibility of a few thinning schedules for spruce stands in Finland. From my understanding, the technical parts of this work were done carefully and correctly, following adequate approaches. I appreciate the effort the author put in the originality of this experience. Therefore, the research question is very interesting. This study provides an important contribution in forest management. The manuscript has significatively improved in comparison with the previous version. For example, the structure is clear now, the methods section is great now, results are well written (in general) and in the discussion the author recognized the study limitations and provide some recommendations.
However, this article has some form weaknesses (moderate and minor in many cases) that could be improved that I list below:
- The introduction did not present the topic topic context, needs... Introduction doesn’t describe the international context. And also, there is some important references lacking. Therefore, half of the introduction only provide methodological information, that should be shown in the methods section.
- The objectives are not clear.
- The Figures must to be improved. Many Figures (13). It must to be merged. Also, the Figures has not enough quality (are excel Figures?).
- The first part of the discussion should be short; the author repeated many results and these paragraphs were written as a report.
- Conclusion is lacking.
I have also provided the following specific suggestions that would help to improve the quality of this manuscript. At this stage, I propose the authors to consider these suggestions in a moderate revision, and request the editor not to accept the manuscript until and unless the authors make the changes. I encourage the author to incorporate these suggestions, and I want to say congratulations for this interesting and original research. The manuscript is reaching the publication, only it is necessary a little bit more of work.
b) Specific comments
Tittle: - The title has improved. However, it is very general yet. I propose to the author to add: the stands (spruce stands) and location (Finland). Could you add more information?: e.g. Haversting design by capital return for spruce stand in boreal forests.
1. Abstract : the major contribution of this study as well as forest management implications in the last two sentences. Conclusion is lacking in the abstract.
2. Keywords: Keywords must to follow alphabetic order. modify it, please. Also, you could add more keywords as growth yield, thinning, financial, sustainable forest management.
3. Introduction:
L22. Problematic and context is lacking. I suggest to add a broad paragraph about thinning, sustainaile forest management in boreal forests and to justify why is important the financial performance.
L22. You start your paragraph with: We are interested… But Why are we interested. ? could you explain it before to present -maximimizing of capital return rate. ?
L22-27: add references.
L33. Add reference.
L35. Why? Provide more information and detail.
L37-40. You present the objective here. This is not a good idea, because you did not provide: background, importance of study it and contribution. I propose you, present the goal at the end of introduction as the classical structure in scientific paper and original research.
41-48: this information should be in the methods section, not in the introduction. Provide references too.
48-55: This paragraph has not important and essential information. Remove it, please. Please, provide a background and justify the importance and contribution of this study. I suggest to add a paragraph to talk about the international context too.
L.43. Could you replace: experimental materials by Experimental design.
Figure 1 and 2: Could you merge in one unique figure. Put the Y tittle on left. Could you add minor ticks? Could you add minor ticks in the x axis, please?
L101-137: please provide more references here about the financial valuations methods.
L56-172. You indicated Fig. 3, three times. With one time is enough. The same for all the manuscript. The same for the other figures in the manuscript.
L170: 5…25. It is strange. Could you reformulate it in other way, please?
Figure 4. Merged in only one Figure the Figure 4, 5, 6 and 7. The x title must to be short!
L186-187. Please, no compare Fig1 with Figure 4. In the results, we describe.. the comparisons must to be in the discussion section. ONLY.
L207. Replace 6 m3 by 6 m3
Figure 7: Describe -a- in the legend.
Figure 8,9,10 and 11: merge it, in only one figure please. Could you add minor ticks in the x axis, please?
Discussion:
L271-336: this part of the discussion is only based on result repetition. Please, could you short it, please. Author refer the Figures all the time, please modify it. The other part of the discussion is great, but form 271-336 is boring. Where is the results comparassion with other studies?
L256 -259. Start the discussion with the main contribution and why this study is important, please.
L270-299. Many results repetition. Add more references
L300-336. Please, could you reduce this section and add more refetences. Where is the results comparassion with other studies?
L302, replace number by the variable … numbers of whats…
347-404: It is fantastic!
347-356: discuss more the discrepancy with these studies, please,
356: Please, replace Fig5 by the concept or study variable. The paper has 11 figures is impossible remember which is the Figure 5.
363: add boreal before regions, please.
L369. Add also this reference here, please:
Montoro Girona M, Rossi S, Lussier J-M, Walsh D, Morin H (2017) Understanding tree growth responses after partial cuttings: A new approach. PLoS ONE 12(2): e0172653. https://doi.org/10.1371/journal.pone.0172653
L377. Add brackets to the Figs. 4,5 and 6).
405: I propose add a paragraph with the conclusion.
Also, there is one study limitations and future recommendations that could be interesting to talk in this paper. Under climate changes disturbances regimes will increase in terms of frequency and severity. However, this study did not take it in to consideration. I propose that author propose to develop more research considering the impact of natural disturbances in the future in the financial feasibility. I propose to write something as: Climate change will alter the natural disturbances regimes (fire, insect pest, wind storms) in terms of frequency and severity in boreal forests (Navarro et al., 2018, Seidl et al., 2014), and it could get a major impact in the financial feasibility. However, in this study the simulation has neglected this factor. Thus, to improve the model estimations we consider to take it into consideration in future researches.
Navarro, L.; Morin, H.; Bergeron, Y.; Girona, M.M. Changes in spatiotemporal patterns of 20th century spruce budworm outbreaks in eastern Canadian boreal forests. Front. Plant Sci. 2018, 9.
Author Response
In the Introduction, the first paragraph states a hypothesis. The second paragraph provides a broad literature review. The third paragraph states the objective of the present study. Obviously, it has not been clear enough. It has now been modified in order to leave no uncertainty. The third paragraph starts with “an improvement harvesting” with indefinite article. Now it is very clearly rewritten in this paragraph what we mean with this concept.
The first four paragrahs of the Introduction have been written in a global context. It is only in the fifth paragraph where a practical focus to a certain localized application is presented. Even in this paragraph an international applicability is discussed, as well as the necessary means for worldwide application.
The Author believes it is customary to provide an outline of the further contents of any paper at the end of the introduction. The Author would like to retain the outline at the end of the Introduction, and present details in the section “Materials and Methods”. The author believes providing an outline at the end of the Introduction makes the content of the paper easier for the readers.
The Author believes he has done his best with the Figures. The total number of Figures has been reduced from 14 to 11. The first two Figures were merged. Unfortunately, it was not possible to merge other Figures, without sacrificing clarity. Two Figures have been removed. The author feels sad about that. The two contained the time-development of basal area and stem count of previously thinned stands. These issues are now discussed on lines 264-273.
The first four paragraphs have been removed from the beginning of section “Discussion”. A new section “Conclusions” has been added after “Discussion”.
In principle, the author understands the difference between “Results” and “Conclusions”. However in this very case he has some difficulty. The first sentence of the Abstract states a hypothesis. Lines 11 and 12 state that the hypothesis showed to be correct. Is that a Conclusion? Yes it is. The rest of the Abstract states condensed results, necessary in the implementation.
In the “Introduction”, hypothesis is stated in the first paragraph. The first paragraph of the section “Conclusions” states that the hypothesis was correct. The rest of “Conclusions” presents condensed results, necessary for implementation. There are references to Figures. If there would not be, I would force any reader to do detective work in order to verify the presented statements. Could I possibly do better?
May I state once more that I would not like to make a specific title for a paper that is intended to be of worldwide importance. There is a practical focus to semiboreal Norway Spruce, but that is not the idea of the content. The idea is:
“Even if the practical focus is in semiboreal spruce stands, the results may be, at least qualitatively, applicable to any circumstances where at least to some degree shade-tolerant trees experience a value-adding assortment transition.”
“In addition to tree species, growth conditions, and models describing growth and recruitment, financial and technical circumstances would vary by geographic region. The basic requirement for the applicability of the results of this study is that trees experience a value-adding assortment transition.”
Would the international focus of the paper possibly be OK?
The Keywords are in alphabetical order.
The second paragraph of the Introduction is a broad review of thinning.
The first sentence of the Introduction has been removed.
The primary objective of the study is to design an improvement harvesting. This is now clarified on lines 35 and 36.
The first paragraph of the Introduction states the hypothesis. The second paragraph presents a literature review related to it. The third paragraph presents the objective. The fourth paragraph concretizes the setup where the objective is to be implemented. The fifth paragraph presents an outline of the remaining content of the paper.
The impression is of this order of things follows a proper logic of an Introduction. The Reviewer obviously does not agree. Fortunately, there is not any one and single correct way of doing things. There are different traditions. The author believes the present arrangement of the Introduction possibly allows any reader to comprehend the contents of the paper.
The paper is intended to reside in an international context. The entire Introduction, is written in an international context, including the fourth paragraph which discusses the practical focus. There, it is stated “Even if the practical focus is in semiboreal spruce stands, the results may be, at least qualitatively, applicable to any circumstances where at least to some degree shade-tolerant trees experience a value-adding assortment transition.”.
Figures 1 and 2 have been merged.
Thank you for pointing out that the subparagraph
2.3. Financial valuations and methods
needs more references. They have now been added.
The reviewer proposed merging rather many Figures. This issue has been investigated very carefully. It was possible to merge only Figures 1 and 2 of the previous manuscript.
The four first paragraphs of the Discussion have been removed.
The comparison of the results with other studies is on lines 359…373.
May I ask why discussions should be restricted to boreal regions?
A paragraph discussing climate change has been added at the end of the Discussion. This issue is rather severe, and I thank the reviewer for pointing this out.
Reviewer 2 Report
Dear Author, I have finished my review on your manuscript. Please refer to the following comments and suggestions: General: Language needs a careful attention Title: Harvesting "planning" by capital return? Abstract: Line 9: whereas others young stand - stands Introduction: Line 29: [1-9] In general, one would like to find an introduction designed to build (a) the problem. I feel that this version willy lacks in doing so. Then, the last paragraph is purely Materials and Methods. I would recommend to improve this section because the readers are interested to see what was the problem before going in detail to see how it was approached. Materials and methods: Subsection 2.1: I think that the data would make more sense if given in a table. Otherwise, in my opinion, it is difficult to understand the data coming from the figure. Also, it is quite common to report the locations of study on a map. Also, I think that subsections should be numbered something like 2.1., 2.2. and so on. Not 21, 22… Not very clear how the model was used in subsection 22. Subsection 23: it is difficult to understand. The author gives some equations and then states that some parameters will not be discussed/used. I strongly suggest to rethink this part to be readable and understandable for the readers since it is not clear what and where was used. Results: In general, the results are very difficult to understand. Take for example lines 177-178. This could be related to the language used as well as to the fact that Materials and Methods failed to clearly present what and how was the study approached; figures need to be self-explanatory. What is a stationary state? What is an improvement harvesting? What is a significant basal area? Define them in Materials and Methods. Line 207: in the order of 30 m3/ha even if the basal area (after harvesting) is only “6 m3/ha”? Discussion: Seems to be OK; Conclusions: Not included.
Author Response
Thank you very much.
I investigated the difference between “Design” and “Planning”.
I found a
difference: “Planning applies established procedures to solve a largely
understood problem within an accepted framework. Design inquires into the
nature of a problem to conceive a framework for solving that problem.” https://fs.blog/2014/12/counterinsurgency-field-manual/
The reference mentioned did agree with my previous conception: Design is a process
of somewhat higher level of abstraction than Planning. My impression is that
the topic of the paper is closer to Design.
On line 9, I am referring to a process “young stand cleaning”. There is no process called “young stands cleaning”. Am I right?
The Reviewer states that the Introduction fails in building up a problem.
It appears that the first paragraph states a hypothesis. The second paragraph provides a literature review. The third paragraph states the objective of the present study. Obviously, it has not been clear enough. It has now been modified in order to leave no uncertainty. The third paragraph starts with “an improvement harvesting” with indefinite article. Now it is very clearly written in this paragraph what is meant with this concept.
Within the science tradition the Author belongs, it is customary to provide an outline of the further contents of any paper at the end of the introduction. The reviewer states the last paragraph belongs to “materials and methods”. However, the paragraph does not discuss any methods in detail. It only provides an outline. The Author would like to retain the outline at the end of the Introduction, and present details in the section “Materials and Methods”. The author feels it is very important to provide the outline at the end of the Introduction, since it is believed to make the paper easier to comprehend.
The Author agrees that Fig. 1 possibly requires more explanation. A new paragraph has been added, explaining the design of Fig. 1. The Author does not think the data would be better described in a Table.
The subsections have now been renumbered as proposed by the Reviewer. The Author agrees that the description of the application of the growth model did rely too much to the references given. Now there is a more comprehensive description.
The reviewer states that in section 2.3 the Author gives equations but then states that some parameters appearing in them would not be discussed. Actually that is not true. All symbols appearing in the equations are actively used in the paper. Two more sentences have been added in order to clarify this.
Regarding the presentation of the results, the Reviewer states they are difficult to understand due to possibly inconvential terminology. She is at least partially correct. The term “Improvement harvesting” has now been introduced in the beginning of the paper, on line 36. The “Stationary state” is introduced on lines 168-169.
The Author thinks it is extremely important to use appropriate terminology. The stationary state discussed here is not any equilibrium state, neither a steady state. These would correspond to vanishing time derivatives of observables. It has been verified from the general science literature that “Stationary state” is the correct term here.
https://arxiv.org/abs/cond-mat/0701683
The concept “basal area” has now been defined at the first instant where it appears, on line 64. And thank you, there was a typo in the unit of basal area on line 219.
Thank you, a new chapter “Conclusions” has been added.
Round 2
Reviewer 1 Report
Congrats for This nice study
This manuscript is a resubmission of an earlier submission. The following is a list of the peer review reports and author responses from that submission.
Round 1
Reviewer 1 Report
a) General comment :
The authors present a study to evaluate the financial feasibility of a few thinning schedules for spruce stands in Finland. From my understanding, the technical parts of this work were done carefully and correctly, following adequate approaches. I appreciate the effort the author put in the originality of this experience. Therefore, the research question is very interesting. This study provides an important contribution in forest management.
However, this article has some weaknesses that could be improved that I list below:
- The title could be improved. It is very general.
- The abstract has not an introduction
- The structure of this manuscript is difficoult to follow and understand.
- The structure of method section must to be improved (see comments below). Also, it is too long.
- The introduction is too short and did not present all the important topics. Introduction doesn’t describe the international context. And also, there is some important references lacking.
- The objectives are not clear.
- Methods must to be re-structured.
Results are written as a report or Master thesis.
- English writing need to be improved:
- The Figures must to be improved. Many Figures (13). It must to be merged. Also, the Figures has not enough quality (are excel Figures?).
- In the discussion, doesn’t describes the international context. Also, in the discussion is lacking the management implication.
I have also provided the following specific suggestions that would help to improve the quality of this manuscript. At this stage, I propose the authors to consider these suggestions in a major revision, and request the editor not to accept the manuscript until and unless the authors make the changes. I encourage the author to incorporate these suggestions, and I want to say congratulations for this interesting and original research.
b) Specific comments
1. Tittle: Could you add more information? E.g. in Finland or in boreal forest. I suggest: Financial feasibility of thinning for spruce stand in boreal forests.
2. Abstract : Could you add an introduction for the abstract? E.g. About thinning. Add the major contribution of this study as well as forest management implications.
3. Keywords: Keywords must to follow alphabetic order.. modify it, please. Also, you could add other keywords as growth yield, partial cutting, sustainable forest management.
4. Introduction:
L20. I suggest to add a broad paragraph about thinning, sustainaile forest management in boreal forests and to justify why is important the financial performance.
L20. You start the two first sentences with the same way: We are interested. Also, in the line 25. Please, modify it!
L29. You present the objective in the second paragraph. This is not a good idea, because you did not provide: background, importance of study it and contribution. I propose you, present the goal at the end of introduction as the classical structure in scientific paper and original research.
L25-30. Any reference here?
L35-41. This paragraph has not important and essential information. Remove it, please. Please, provide a background and justify the importance and contribution of this study. I suggest to add a paragraph to talk about the international context too.
Methods: this section is difficult to understand because the structure. Author must to provide subsections.
L.43 Add a new subsection here: Experimental design.
L43: replace 11 by Eleven.
L43. When you say: spruce dominated forest… Please add the scientific name: Norway spruce…
43-47: Could you add a map with the study sites location?
L43-47: Describe what’s thinning. Add the harvest intensity, and the silvicultural prescription, the types of thinning. When they were performed.
Figure 1 and 2: Merged in one unique figure. Put the Y tittle on left. Use only one legend.
48-69: You show the stand selection? If yes please add a new Subsection: stand selection. You star both paragraph with: we find from Fig.1 or 2. Please, change it. This is a scientific paper not a report.
L66: No list please…
L76… Could you add a new section: Growth and financial models or data analyses.
L118: replace Euros by the symbol €
L123, 124 and 130. You indicated Fig. 3. With one time is enough. The same for all the manuscript.
L126: no list, please.
L.154. Delete Evolution from the Legend.
Figure 4. Merged in only one Figure the Figure 4, 5, 6 and 7. The x title must to be short!
L160. … 170 ? ha… Could you indicate the unit please?
L164-165. Please, no compare Fig1 with Figure 5. In the results, we describe.. the comparisons must to be in the discussion section. ONLY.
Figure 7: (5a ha) what’s it? Describe it in the legend.
L. 186. .Remove space before The
L196. Remove space before AGAIN.
L204. I don’t understand: 1.7 … 5.0. Also, replace , by .
L.230. Use past form, please.
To build a new Figure with the Figure 8, 9, 10, 11, 12 and 13.
Discussion:
L256 -259. Start the discussion with the main contribution and why this study is important.
L256. Stands discussed below… report style! Re-write, please..
L260-272. Many results repetition. Add more references
L273. Define what’s young stands or add the age and one reference.
L.276. High thinnings, please add harvest intensity (). And a reference.
L281. 31,,,,45 modify it.
L278-286. Add references.
L351. You could add references for other spruce species and to difereciate between European and North-American boreal forests.
In some parts of the discussion you talk about artificial regeneration, however natural regeneration is lacking.
a) Study limitations and future recommendations:
a. The authors used a empirical model constructed with Norwegian data set. Maybe the model parameters and the resolution is not the best for Finish species, based on growth is different from ecoregions.
b. Authors must to do recommendations for future researches. Also, authors have analysed it using stand models. Authors did not consider the individual tree variability. I suggest that author said in the discussion: - For future researches, we recommended to explore the growth response using individual non-linear models to get a better resolution of growth (Montoro Girona et al., 2017). Reference: Montoro Girona M, Rossi S, Lussier J-M, Walsh D, Morin H (2017) Understanding tree growth responses after partial cuttings: A new approach. PLoS ONE 12(2): e0172653. https://doi.org/10.1371/journal.pone.0172653
b) International context is lacking: A small paragraph to discuss other studies in other forest ecosystem could be great for the readers and to improve the Scopus of this paper. For examples. Author could give recommendations for future research in Canadian boreal forests. I suggest to indicate something as: Recently, new shelterwood silvicultural treatments has been developed in black spruce stands from Canadian boreal forest. These treatments showed a high growth response on residual trees, intermediate level of mortality for wind damages, as well as provided adequate level of natural regeneration (Montoro Girona, 2016, 2018). Based on the results obtained here, we recommended to study the financial feasibility of these treatments. References:
Montoro Girona, M.; Morin, H.; Lussier, J.-M.; Walsh, D. Radial Growth Response of Black Spruce Stands Ten Years after Experimental Shelterwoods and Seed-Tree Cuttings in Boreal Forest. Forests 2016, 7, 240.
Montoro Girona, F. M. (2017). À La Recherche De L'aménagement Durable En Forêt Boréale: Croissance, Mortalité Et Régénération Des Pessières Noires Soumises À Différents Systèmes Sylvicoles. Ph.D. thesis, University of Quebec in Montreal.
Montoro Girona, M., Morin, H. H., Lussier, J.-M. J., and Thiffault, N. N. (2018a). Conifer regeneration after experimental shelterwood and seed-tree treatments in boreal forests: finding silvicultural alternatives. Front. Plant Sci. 9:1145. doi: 10.3389/fpls.2018.01145
Reviewer 2 Report
Dear Author(s),
The paper presents an interesting case study for the conditions of Finland. However, it is limited in scope to an international readership and it could fit better the requirements of a local journal where the presented kind of forest-management system is applied. Therefore, I would recommend the author to prepare and submit the paper to a specialized local journal of forestry.
To be more specific,
1.) Title is too general for what has been reported in the paper. Spruce stands are characteristics to many regions but the management of these is different;
2.) Not sure that the Abstract is formatted according to the requirements of the journal;
3.) Introduction section is lacking a clear description of the problem and it seems to be insufficiently documented. Rather it reflects a personal opinion of the author followed by a mixture of text that should be placed in Materials & Methods, with no natural flow that should present to the reader the gap of knowledge and the transition to the problem to be solved followed by the aim and objectives of the paper;
4.) Materials and Methods section seems to be inappropriately formatted with unorganized information that is difficult to digest and read. For this section I would recommend a more systematic approach to get some sense for the readers; this could be achieved by including some description of the stands, placement in coordinates or a map, placing some descriptive tables and referencing the methods used to that already existing. Also, the format of the text, English spelling and terminology used should be carefully checked;
5.) Results section lacks a clear and systematized description. There is a bunch of figures that should have some more descriptive legends, and probably most of the result could be systematized by using more compressed means of reporting.
6.) Discussion may be appropriate, but it is, still, based on a large number of Finnish studies, therefore indicating the local nature of results.
7.) References: the author used predominantly Finnish references that probably suit the needs of his study but lack in the general audience when comparing the results.
In general,
The manuscript could be scientifically sound but it is hard to say if that’s true due to an improper organization. The readers will be confused in many cases because they will find some repetitive information and not the kind of text organization they are expecting. English spelling should be checked in detail and the correct terms should be used.